# Salient experiences are represented by unique transcriptional signatures in the mouse brain

Diptendu Mukherjee[1], Bogna Marta Ignatowska-Jankowska[2], Eyal Itskovits[3,4], Ben Jerry Gonzales[1], Hagit Turm[1], Liz Izakson[1], Doron Haritan[1], Noa Bleistein[1], Chen Cohen[1], Ido Amit[5], Tal Shay[6], Brad Grueter[7], Alon Zaslaver[3], Ami Citri[1,2,8]*

[1]Department of Biological Chemistry, Silberman Institute of Life Sciences, The Hebrew University, Jerusalem, Israel; [2]The Edmond and Lily Safra Center for Brain Sciences, The Hebrew University, Jerusalem, Israel; [3]Department of Genetics, Silberman Institute of Life Sciences, The Hebrew University, Jerusalem, Israel; [4]School of Computer Science and Engineering, The Hebrew University, Jerusalem, Israel; [5]Department of Immunology, Weizmann Institute of Science, Rehovot, Israel; [6]Department of Life Sciences, Ben-Gurion University of the Negev, Beer-Sheva, Israel; [7]Department of Psychiatry, Vanderbilt University School of Medicine, Nashville, United States; [8]Child and Brain Development Program, Canadian Institute for Advanced Research, Toronto, Canada

**Abstract** It is well established that inducible transcription is essential for the consolidation of salient experiences into long-term memory. However, whether inducible transcription relays information about the identity and affective attributes of the experience being encoded, has not been explored. To this end, we analyzed transcription induced by a variety of rewarding and aversive experiences, across multiple brain regions. Our results describe the existence of robust transcriptional signatures uniquely representing distinct experiences, enabling near-perfect decoding of recent experiences. Furthermore, experiences with shared attributes display commonalities in their transcriptional signatures, exemplified in the representation of valence, habituation and reinforcement. This study introduces the concept of a neural transcriptional code, which represents the encoding of experiences in the mouse brain. This code is comprised of distinct transcriptional signatures that correlate to attributes of the experiences that are being committed to long-term memory.

DOI: https://doi.org/10.7554/eLife.31220.001

*For correspondence: ami.citri@mail.huji.ac.il

## Introduction

Neuronal plasticity enables cognitive and behavioral flexibility underlying the development of adaptive behaviors (*Alberini, 2009*; *Alberini and Kandel, 2015*). This neuroplasticity, induced by salient experiences, has been shown to depend on the induction of temporally-defined waves of transcription (*Alberini, 2009*; *Alberini and Kandel, 2015*; *McClung and Nestler, 2008*; *Flavell and Greenberg, 2008*; *West and Greenberg, 2011*). The earliest of these waves consists of the expression of immediate-early genes (IEGs). IEGs have been conventionally treated as molecular markers for labeling neuronal populations that undergo plastic changes during the formation of long-term memory (*Cruz et al., 2013*; *Minatohara et al., 2015*). However, the literature indicates a much more significant contribution of IEGs to synaptic plasticity and memory formation (*Lanahan and Worley, 1998*; *Okuno, 2011*). It has been proposed that IEG transcription may represent the molecular signatures

**eLife digest** Can we tell what important event a mouse – or even a person – has recently experienced? The current experience of an individual can be inferred from brain imaging experiments. However, along with changing brain activity, such an experience also switches on gene activity throughout the brain. This enables neurons to produce the proteins required to form a long-term memory of the experience.

Do distinct, memorable experiences trigger unique signatures of gene activity? To answer this question, Mukherjee, Ignatowska-Jankowska, Itskovits et al. exposed mice to a variety of experiences. Some were unpleasant and induced aversion; for example, the mouse may have felt nauseous or experienced brief pain and fear. Other experiences, such as when the mouse drank sugary water, received food or was injected with cocaine, were rewarding.

Each of the experiences led to the activation of unique combinations of genes in different regions of the brain. Analysing a subset of the activated genes in various brain regions led to the identification of unique and reliable gene expression signatures of experience. These signatures allowed the recent experience of mice to be decoded with nearly 100% accuracy. While these unique signatures can distinguish between recent experiences, experiences that share common features do trigger overlapping patterns of gene activation. For example, negative experiences – but not positive or neutral ones – activated similar patterns of genes in a brain region called the amygdala. In contrast, repeated rewarding experiences induced a distinct gene activity pattern that was most pronounced as increased activity in part of the brain called the frontal cortex.

These findings increase our understanding of how the brain represents information. The approach described in the paper provides a strategy to measure the changes in the brain that occur when information is encoded for long-term storage. This measure could also be useful during drug development, revealing how new drug compounds affect the brain, as well as providing an objective measure of the subjective experience of an individual. For example, substances that trigger similar patterns of gene activation to addictive drugs may themselves be addictive. On the other hand, substances that induce similar activity patterns to known medications could also have similar therapeutic properties.

DOI: https://doi.org/10.7554/eLife.31220.002

of long-term plastic changes underlying the formation of memory (*Alberini, 2009*). Thus, induced IEG transcription could represent an experience-specific neural code for long-term storage of information. The existence of a neural code embedded in transcription implies that it should be possible to decode the identity of recent experiences, and potentially derive information regarding the nature of the experience, from its transcriptional representation (*Stanley, 2013*).

To address the existence of a neural transcriptional code, we performed a detailed analysis of IEG transcription for 13 different experiences: cocaine (acute, repeated and challenge), volitional sucrose drinking (acute and repeated), reinstatement of feeding following food deprivation, lithium chloride administration (LiCl; acute and repeated), saline (acute injection without habituation, acute injection after habituation and repeated administration), acute administration of a mild foot shock, and exposure to a novel chamber with no foot shock. The experiences were selected to enable the identification of the transcriptional representations of affective attributes, such as salience and valence (*Russell, 1980*; *Posner et al., 2005*). As such, we chose to investigate experiences that drive robust positive or negative reinforcement. Repetition of rewarding and aversive experiences provided insight into the transcriptional representation of habituation to negative stimuli and positive reinforcement of rewarding experiences.

Experiences included in this study have been previously studied using electrophysiological measures, and plasticity has been observed within individual limbic and mesolimbic brain structures (*Russo and Nestler, 2013*). In contrast to classic electrophysiological measurements of plasticity, which focus on measurements with synapse specificity, transcriptional analysis enables parallel investigation of the representation of experience across multiple brain structures. Assuming that the encoding of complex reinforced experiences involves coordinated neural plasticity in multiple brain regions, we analyzed transcription across structures associated with the limbic and mesolimbic

systems (*Russo and Nestler, 2013*; *Haber and Knutson, 2010*). The brain structures that were analyzed include limbic cortex (LCtx; including medial prefrontal cortex and anterior cingulate cortex), nucleus accumbens (NAc), dorsal striatum (DS), amygdala (Amy), lateral hypothalamus (LH), dorsal hippocampus (Hipp) and ventral tegmental area (VTA).

Our results demonstrate that the transcriptional representations of experience are robust, reliable and consistent, enabling the decoding of the recent experience of mice with high levels of accuracy from a minimal transcriptional signature. We identify transcriptional hallmarks of affective attributes of experience, prominently demonstrated in the encoding of valence. Moreover, we report opposing patterns of transcriptional modulation underlying the development of habituation to experiences of negative valence, in comparison to repeated rewarding experiences associated with positive reinforcement. We conclude with a discussion of the potential implications of a neural transcriptional code.

## Results

### Identification of transcriptional signatures of experience

We initiated our study with the investigation of gene expression programs induced during the development of behavioral sensitization to cocaine. Cocaine sensitization is one of the most widely applied paradigms for studying mechanisms of neural plasticity, due to the robustness of the behavioral model and the detailed insight acquired into the underlying mechanisms (*McClung and Nestler, 2008*; *Robbins et al., 2008*; *Hyman et al., 2006*; *Nestler, 2002*; *Robison and Nestler, 2011*; *Lüscher, 2016*; *Piechota et al., 2010*). Using this paradigm, we studied the transcriptional programs induced following acute or repeated exposure to cocaine, as well as re-exposure to cocaine after a period of abstinence from repeated drug exposures ('cocaine challenge') (*Figure 1A,B*) (*Robison and Nestler, 2011*). We analyzed the transcription induced at 0, 1, 2, 4 hr following each of these cocaine experiences across six brain structures (LCtx, NAc, DS, Amy, LH, and Hipp; *Figure 1—figure supplement 1*). Transcription was analyzed with a comprehensive set of qPCR probes against putative IEGs (see Materials and methods and *Supplementary file 1*). Our results demonstrate that distinct cocaine experiences (acute, repeated, challenge) are characterized by robust induction of a handful of genes across the different brain structures studied, with peak induction at 1 hr following cocaine administration (*Figure 1*; *Figure 1—figure supplement 2*; transcriptional dynamics shown in *Figure 1—figure supplement 3*). 29 genes were induced above two fold in at least one of the six brain regions (predominantly in LCtx, NAc and DS), across the three cocaine experiences.

We were next interested in comparing the transcription programs induced by cocaine with those induced by an experience of opposite valence, within the same experimental context. For this purpose, we performed acute, as well as repeated, administration of the pharmacological compound LiCl, which induces malaise and decreases locomotion (*Figure 1D,E*)(*Fortin et al., 2016*). Similar to cocaine, LiCl drove robust induction of a small subset of IEGs (*Figure 1F*). In the case of LiCl experiences, 30 genes were induced above 2-fold, predominantly in the LCtx, Amy and LH (*Figure 1*, *Figure 1—figure supplement 4*; two doses of LiCl (150/250 mg/kg) induced transcriptional responses of similar magnitude - *Figure 1—figure supplement 4*). As a reference for the transcription induced by cocaine and LiCl experiences, we characterized the transcription induced by saline in control animals (before and after habituation, as well as following repeated exposure; *Figure 1—figure supplements 5* and *6*). Cocaine and LiCl experiences shared a common core set of 16 genes (*Arc, Atf3, Cyr61, Dusp1, Egr2, Egr4, Elovl1, Enpp6, Fos, Fosb, JunB, Ier2, Ier5, Nr4a1, Ngfr* and *Npas4*) of which we selected five genes for further investigation. Marker gene selection was performed by ranking genes according to the frequency of their induction (i.e. # of appearances above two-fold induction from 30 possible appearances across 6 structures in five experiences), as well as ranking in inverse order of the average variance ($S^2$) of the magnitude of induction. The five genes with combined highest ranking in frequency of appearance and lowest variance in expression were selected for further analysis (*Arc, Egr2, Egr4, Fos* and *Fosb*; *Figure 1G*).

To test our hypothesis that experiences can be decoded from patterns of induced transcription, mice were classified based on the induction of five genes (*Arc, Egr2, Egr4, Fos* and *Fosb*) across five brain structures (LCtx, NAc, DS, Amy and LH), defining 25 gene-structure 'features'. Classification

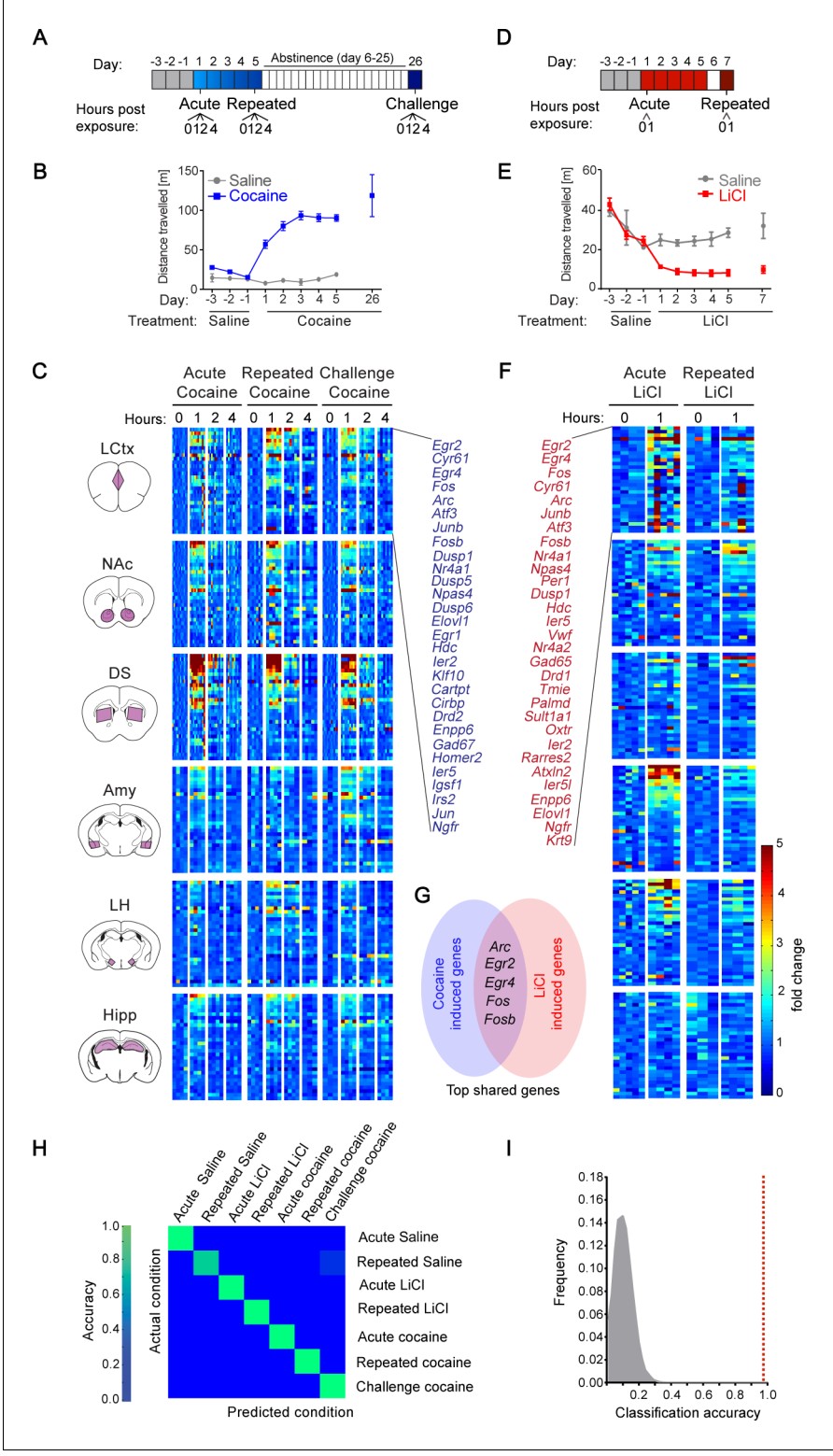

**Figure 1.** Transcriptional signatures representing recent experience. (**A**) Schematic of experimental paradigm for cocaine sensitization. Mice were exposed to cocaine (i.p., 20 mg/kg) or saline, either acutely, repeatedly or re-exposed following abstinence (challenge), with transcriptional dynamics studied at 0, 1, 2 or 4 hr. (**B**) Locomotor activity of mice following acute, repeated or challenge cocaine experiences (compared to saline). Sample size: acute and repeated saline n = 4; acute cocaine n = 30; repeated cocaine n = 22; challenge cocaine n = 19 mice. Results indicate mean ± s.e.m. (**C**) Expression matrix of IEG induction dynamics following cocaine experiences.

*Figure 1 continued on next page*

*Figure 1 continued*

Individual animals are represented in columns sorted according to time points of cocaine experiences [sample numbers per time point - LCtx: limbic cortex (n = 5–11), NAc: nucleus accumbens (n = 5–12), DS: dorsal striatum (n = 5–12), Amy: amygdala (n = 3–4), LH: lateral hypothalamus (n = 2–4), Hipp: hippocampus (n = 2–4)]. Fold induction is graded from blue (low) to red (high). Genes represented were induced at least 2-fold over control in any one of the brain regions studied. Genes were sorted according to peak induction in the DS. (D) Schematic of experimental paradigm for LiCl exposure. Mice were exposed to LiCl (i.p.) or saline, either acutely (150 or 250 mg/kg) or repeatedly (150 mg/kg). (E) Locomotor activity of mice following acute or repeated LiCl exposure (as in panel C). Sample size: n = 4–5. (F) Expression matrix of IEG induction dynamics following LiCl experiences (n = 4–5). Genes were sorted according to peak induction in the Amy. (G) Venn diagram indicating the identity of the most robustly induced genes common to cocaine and LiCl experiences (most appearances and lowest variance). (H) Confusion matrix representing the classification accuracy of decoding the recent experience (acute, chronic and challenge cocaine, acute and repeated LiCl and acute and repeated saline) of individual mice based on expression of *Arc, Egr2, Egr4, Fos* and *Fosb* induction in the LCtx, NAc, DS, Amy and LH using a KNN classifier. Accuracy is scaled from blue to green, with bright green corresponding to 100% accuracy (n = 37 mice). Overall accuracy = 97.3%. (I) Results of a permutation test for verifying classification. A randomization test was performed, in which the classifier was run on $10^5$ random permutations of the association of individual mice to the appropriate experience, and the frequency of classification accuracies is plotted in grey, while the red dotted line represents the classification accuracy obtained for non-randomized data (97.3%).

DOI: https://doi.org/10.7554/eLife.31220.003

The following figure supplements are available for figure 1:

**Figure supplement 1.** Boundaries of dissected structures.
DOI: https://doi.org/10.7554/eLife.31220.004

**Figure supplement 2.** Acute, repeated and challenge cocaine experiences induce distinct transcriptional programs across brain structures.
DOI: https://doi.org/10.7554/eLife.31220.005

**Figure supplement 3.** Time course of transcriptional induction of a minimal set of markers representing cocaine-induced transcriptional dynamics.
DOI: https://doi.org/10.7554/eLife.31220.006

**Figure supplement 4.** Transcriptional representation of LiCl experience.
DOI: https://doi.org/10.7554/eLife.31220.007

**Figure supplement 5.** Characterization of repeated saline exposures illustrates the effect of habituation on induced transcription.
DOI: https://doi.org/10.7554/eLife.31220.008

**Figure supplement 6.** Characterization of the transcription induced by acute and repeated saline experiences.
DOI: https://doi.org/10.7554/eLife.31220.009

performed according to these 25 features using the k-Nearest Neighbors algorithm (KNN) allowed precise allocation of individual animals based on the identity of the recent experience with 97.3% accuracy, such that only one mouse (out of 37) was incorrectly classified (*Figure 1H*).

Taken together, these results suggest that induced transcriptional signatures, defined by the combinatorial expression of minimal subsets of IEGs across brain structures, can be derived from comprehensive gene expression programs induced following an experience. Moreover, these minimal subsets are sufficient to decode the recent salient experience of mice.

## Distinct experiences are represented by unique transcriptional signatures

To further address the existence of a transcriptional code for experience, we expanded our study, including naturalistic volitional experiences of positive valence – sucrose consumption and reinstatement of feeding, as well as foot shock, an additional experience of negative valence. To provide a birds-eye view of the transcriptional landscape, we represent the experience-specific transcriptional signatures induced by each of these experiences using radar plots (*Figure 2*). This representation enables immediate identification of the major transcriptional attributes of each of the experiences. Four genes (*Arc, Egr2, Egr4* and *Fos*) are shown for simplicity of presentation; for individual mice, see *Figure 2—figure supplement 1*. For full data, see *Supplementary file 2*.

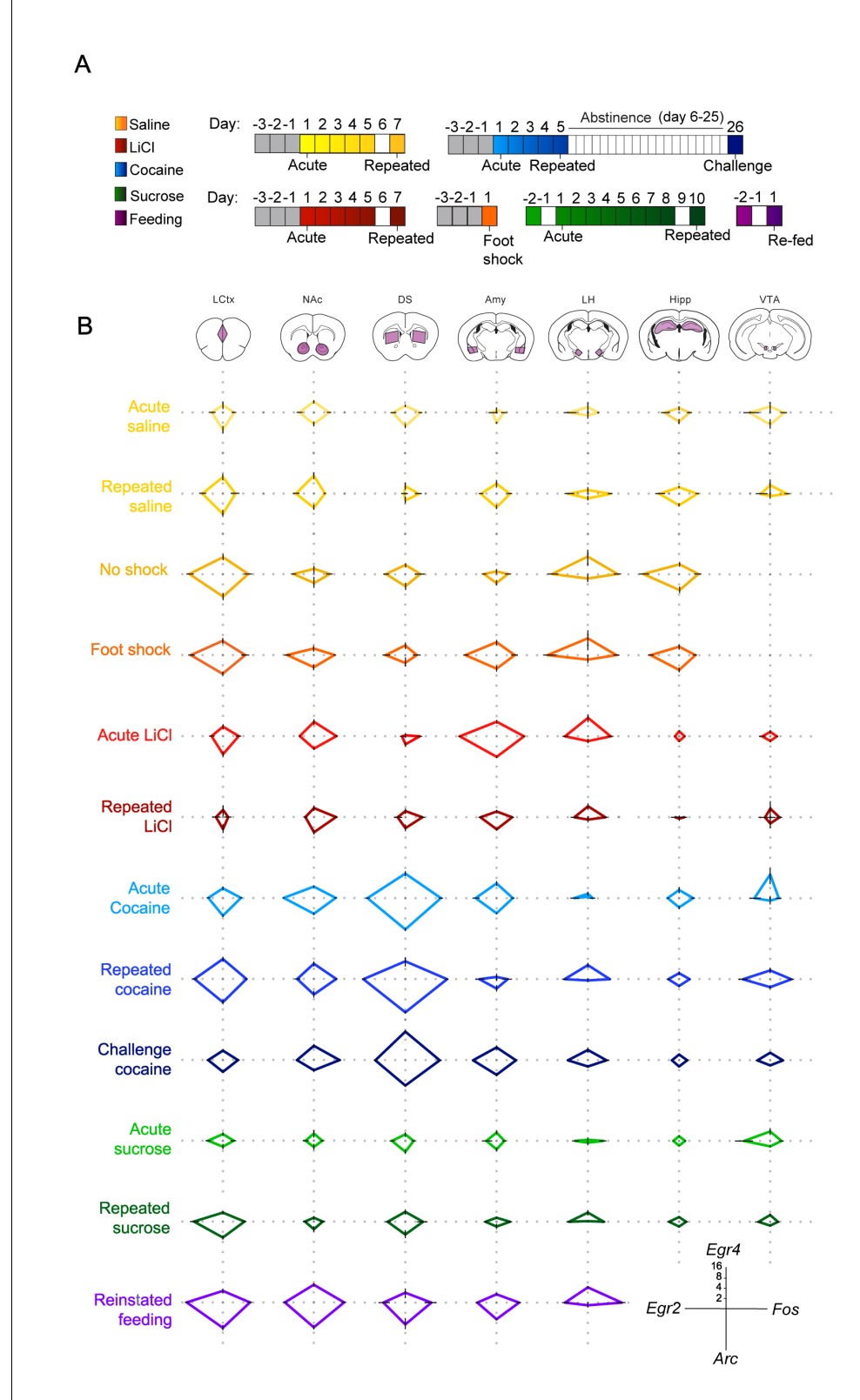

**Figure 2.** Salient experiences are represented by unique transcriptional signatures. (**A**) Schematic of experimental paradigms. Experiences analyzed include saline (acute and repeated); foot shock (acute shock and no-shock controls exposed to the same environment); LiCl (acute and repeated); cocaine (acute, repeated and challenge following abstinence); sucrose (acute and repeated) and reinstatement of feeding (following 18 hr of deprivation).
*Figure 2 continued on next page*

*Figure 2 continued*

(B) Radar plots representing the transcriptional induction of *Arc, Egr2, Egr4* and *Fos* across seven brain structures 1 hr after the different experiences [LCtx: limbic cortex (n = 4–14), NAc: nucleus accumbens (n = 4–14), DS: dorsal striatum (n = 4–14), Amy: amygdala (n = 4–9), LH: lateral hypothalamus (n = 3–9), Hipp: hippocampus (n = 4–9); VTA: ventral tegmental area (n = 2–8)]. Results are shown in $\log_2$ scale as mean ± s.e.m. of induction over baseline control.
DOI: https://doi.org/10.7554/eLife.31220.010

The following figure supplements are available for figure 2:

**Figure supplement 1.** Low variability of the individual transcriptional representations of recent experience.
DOI: https://doi.org/10.7554/eLife.31220.011

**Figure supplement 2.** Transcriptional representation of negative valence in the amygdala.
DOI: https://doi.org/10.7554/eLife.31220.012

**Figure supplement 3.** Transcriptional representation of habituation and reinforcement.
DOI: https://doi.org/10.7554/eLife.31220.013

**Figure supplement 4.** Reinstatement of feeding is represented by robust transcriptional dynamics.
DOI: https://doi.org/10.7554/eLife.31220.014

**Figure supplement 5.** Reinstatement of feeding is represented by robust transcriptional dynamics.
DOI: https://doi.org/10.7554/eLife.31220.015

This presentation further highlights the unique nature of the transcriptional signatures characterizing each experience, and the dynamic changes in IEG induction following repeated experience. Furthermore, commonalities in the transcriptional representation of experiences with shared affective attributes are visually apparent in this presentation.

## Transcriptional representation of positive and negative valence

To investigate the transcriptional representation of negative valence, we focused on the aversive experiences induced either pharmacologically by LiCl administration, or by acute administration of mild foot shock. It is worth noting that while LiCl and foot shock are both characterized by negative valence, they are otherwise distinct; LiCl causes visceral discomfort and reduced locomotion (*Fortin et al., 2016*), while foot shock induces acute pain and fear (*Bali and Jaggi, 2015*). Interestingly, exposure to the experimental context (a 18 × 20 cm perspex chamber with a metal grid floor) was by itself sufficient to induce IEG transcription across multiple structures in naïve mice ('no shock' control; *Figure 2*). Mice that received a foot shock within this context displayed an indistinguishable pattern of transcriptional induction compared to their 'no shock' controls (*Figure 2—figure supplement 2*, *Supplementary file 3* - T4), with the sole distinction being a robust induction of transcription in the Amy (primarily of *Egr2* and *Egr4*; *Figure 2*, *Figure 2—figure supplement 2*, Statistics *Supplementary file 3* – T4, Row 4 – Columns B, C). This result demonstrates transcriptional coding of negative valence in the Amy, induced by the addition of a single variable (foot shock) to the experience of exposure to a novel environment. This observation was supported by the transcriptional representation of acute LiCl, which drove induction of *Arc, Egr2* and *Fos* in the Amy (*Figure 2*, *Figure 2—figure supplement 3*, Statistics *Supplementary file 3* – T3, Row 4 – Columns A, B, D).

In contrast to the experiences of negative valence, the rewarding experiences of cocaine, sucrose and feeding had a broader representation across brain structures, which was most obvious in the case of feeding, where significant gene induction was observed across all structures studied (*Figure 2—figure supplements 4* and *5*; Statistics *Supplementary file 3* – T6). The representation of acute cocaine was primarily observed in striatal regions (DS and NAc) and small but significant changes were also observed in additional mesocorticolimbic structures (VTA, Amy, LCtx; *Figure 2—figure supplement 3*; Statistics *Supplementary file 3* – T2), while the representation of acute sucrose was quite minimal, and was reinforced upon additional exposure, as discussed below (*Figure 2—figure supplement 3*; Statistics *Supplementary file 3* – T5).

## Opposing trajectories of the representation of repeated positive and negative experiences

Repetition of aversive or rewarding experiences drove opposing trajectories of IEG induction (*Figure 2*; *Figure 2—figure supplement 3*). Following repeated exposure to LiCl, we observed a

significantly diminished transcriptional representation in the Amy, to levels similar to those observed following repeated saline experience [interaction of treatment (LiCl vs saline) and time (acute vs repeated); *Egr2*: $F_{(1,18)}$ = 8.47, p<0.01; *Fos*: $F_{(1,20)}$ = 17.2, p=0.001, *Arc*: $F_{(1,20)}$ = 8.72, p<0.01] (Statistics *Supplementary file 3* – T3 – row 4). In contrast, repeated exposure to cocaine administration was associated with enhanced transcriptional induction in the LCtx, DS, and VTA (Statistics *Supplementary file 3* – T2 – rows 1,3,7). This enhancement was characterized by the significant induction of *Egr2* in the LCtx and DS and *Fos* in the LCtx, DS, VTA [interaction of treatment (cocaine vs saline) and time (acute vs repeated); *Egr2*: LCtx $F_{(1,29)}$ = 6.43, p<0.05; DS $F_{(1,29)}$ = 4.58, p<0.05; *Fos*: LCtx $F_{(1,29)}$ = 5.35, p<0.05; DS $F_{(1,29)}$ = 4.21, p<0.05, VTA $F_{(1,13)}$ = 14.3, p<0.01] (Statistics *Supplementary file 3* – T2 – rows 1,3,7 columns B,D). However, in the NAc, the initially robust induction of *Egr2* transcription following acute cocaine decreased after repeated administration (interaction of treatment and time, *Egr2*: F (1, 28)=39.7, p<0.0001) (*Figure 2*, *Figure 2—figure supplement 3*; Statistics *Supplementary file 3* – T2 – Column B row 2).

Repeated exposure to sugar was also represented by significantly enhanced transcription, most prominently in the LCtx [interaction of sucrose (sucrose vs water) and time (acute vs repeated); *Egr2*: $F_{(1,26)}$ = 5.02, p<0.05, *Fos*: $F_{(1,26)}$ = 7.51, p=0.01; *Arc*: $F_{(1,26)}$ = 6.79, p<0.05] (*Figure 2*, *Figure 2—figure supplement 3*; Statistics *Supplementary file 3* – T5 – row 1, columns A,B,D). Furthermore, reinstatement of feeding was also represented by significant induction of IEGs in the LCtx, specifically *Egr2* and *Fos* (*Egr2*: $F_{(2,28)}$ = 13.1, p<0.0001; *Fos*: $F_{(2,31)}$ = 41.5, p<0.0001) (*Figure 2*, *Figure 2—figure supplements 4* and *5*; Statistics *Supplementary file 3* – T6 – row 1, columns B,D). The experiences of repeated cocaine, repeated sucrose and reinstatement of feeding, though quite diverse in many affective and cognitive aspects, are all characterized by positive valence and therefore positive reinforcement. Our results suggest that a hallmark of increasing salience of positively reinforcing experiences may be increased transcriptional representation, specifically in the LCtx (*Robinson and Berridge, 2008*). This transcriptional representation of positively reinforcing experiences contrasts with the diminished transcriptional representation associated with habituation to anticipated and unavoidable aversive experiences.

## Decoding recent experiences of individual mice from minimal transcription

Finally, we tested our capacity to decode the recent experience of mice on the full complement of experiences studied in this project. The transcriptional induction of five genes (*Arc, Egr2, Egr4, Fos and Fosb*) across five structures (LCtx, NAc, DS, Amy, LH) forms 25 gene-structure 'features', which were used for the decoding with the KNN algorithm. We found that these 25 features supported the decoding of the recent experience of individual mice with 90.7% efficiency (*Figure 3A*). Random shuffling of the association of mice to experiences demonstrated the reliability of the classifier, and the potential for our results to generalize beyond the given dataset (p<$1e^{-5}$; *Figure 3B*). These results suggest that obtaining a reliable transcriptional representation of a recent experience requires knowledge regarding both the transcriptional induction of several genes and the identity of structures within which they are induced. To further test this hypothesis, we ran a number of permutations. We tested the capacity to decode recent experiences following averaging the data for each gene across the five tested structures (losing spatial information; *Figure 3—figure supplement 1*; classification accuracy 55%), as well as decoding by individual structures (the expression of 5 genes in a single structure; *Figure 3C*; classification accuracies 33–56%) or individual genes (the expression of a single gene across five structures; *Figure 3D*; classification accuracies 37–70%). Taken together, while we find that measurement of the expression of individual genes, such as *Fos* and *Egr2*, across the five brain structures can support classification (67%, 70% respectively), the prediction is significantly improved by the measurement of multiple features (*Figure 3A*).

With the objective of identifying the individual features that provide maximal support for decoding, we performed Random K-Nearest Neighbor (R-KNN) feature selection (*Figure 3—figure supplement 2A*) (*Li et al., 2011*). We identified that a combination of eight features (expression of *Egr2* and *Fos* in the LCtx, NAc and Amy, and expression of *Egr2* and *Fosb* in the DS) provided the highest support, with a decoding efficiency of 93.6% (*Figure 3—figure supplement 2B,C*). An independent approach for feature selection (Breiman Random Forest [*Breiman, 2001*]) identified a largely overlapping set of features, with the top 10 features supporting a classification accuracy of 94.4% (p<$1e^{-5}$; *Figure 3—figure supplement 2E–H*). An intuitive representation of the divergence of

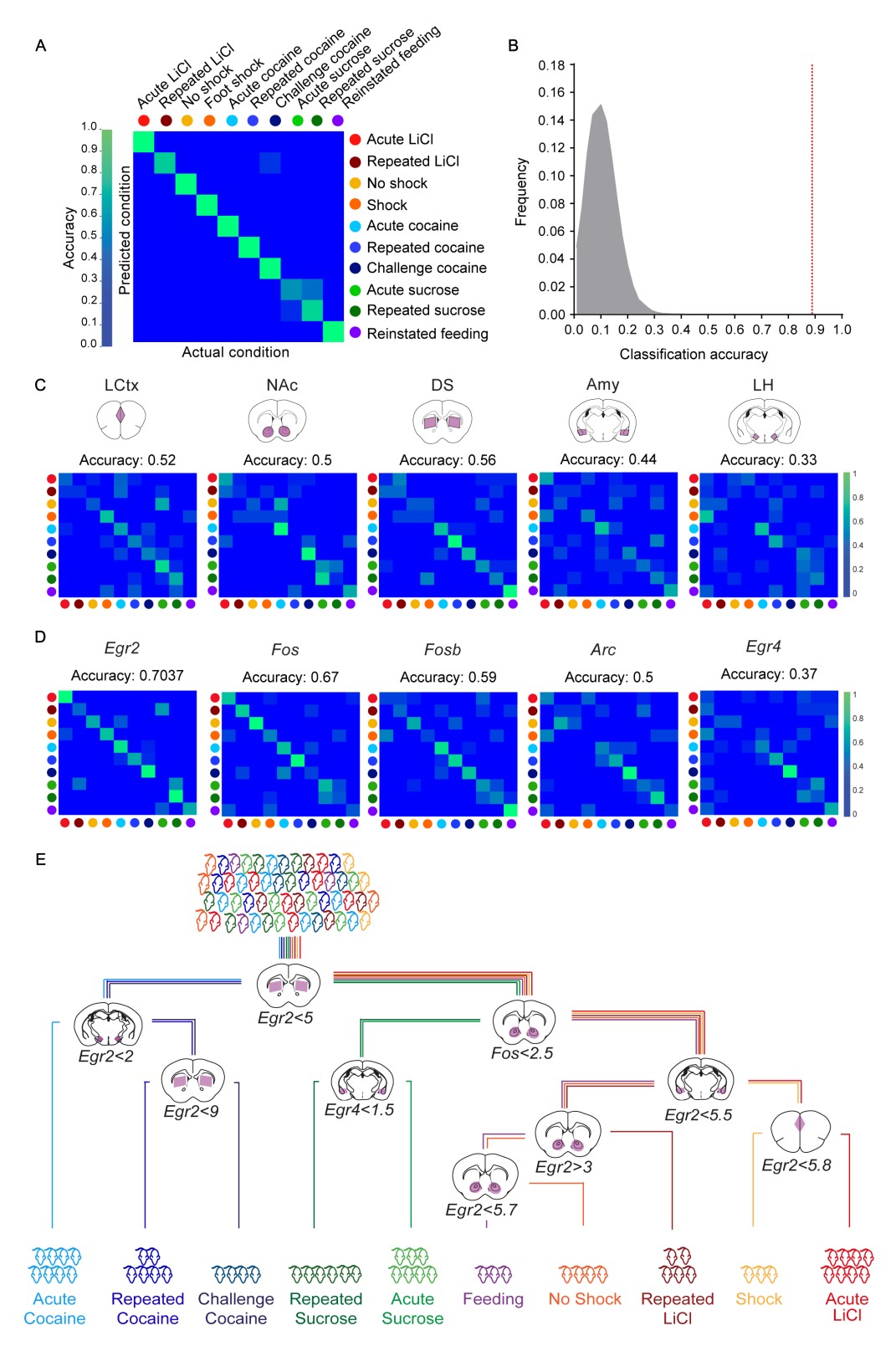

**Figure 3.** Decoding the recent experience of individual mice from minimal transcriptional signatures. (**A**) Confusion matrix representing the classification accuracy (90.7%) of decoding the recent experience of individual mice based on 25 features. Efficiency is scaled from blue to green, with bright green corresponding to 100% efficiency (n = 54 mice). (**B**) Verification of classification validity. A randomization test was performed, in which the classifier was run on $10^5$ random permutations of the association of individual mice to the appropriate experience, and the frequency of classification

*Figure 3 continued on next page*

*Figure 3 continued*

accuracies is plotted in grey, while the red dotted line represents the classification accuracy obtained for non-randomized data (90.7%). (**C, D**) Confusion matrices representing the classification accuracy of decoding utilizing transcriptional measurements from individual brain structures (five genes in one structure, (**C**) or individual genes (single genes across five structures, (**D**). Dots represent the identity of the experience, color-coded according to A. X and Y axes denote the actual and predicted conditions. (**E**) A decision tree enabling the classification of mice according to experience by minimal gene expression (one of many possible trees which can equivalently segregate the data). Mice are classified based on features that enable maximal segregation at each internal node. The thresholds define the allocation of mice to the left branch of the tree at each bifurcation. Mice are color-coded according to experience.

DOI: https://doi.org/10.7554/eLife.31220.016

The following figure supplements are available for figure 3:

**Figure supplement 1.** Decoding the recent experience of individual mice from averaged gene expression across different brain regions.
DOI: https://doi.org/10.7554/eLife.31220.017

**Figure supplement 2.** Feature selection to identify the features contributing most significantly to decoding.
DOI: https://doi.org/10.7554/eLife.31220.018

experiences based on particular features is provided by a decision tree (one of a number of possible trees), in which mice were assigned to appropriate branches according to the extent of induction of a particular gene in a given structure (*Figure 3E*).

Taken together, these results establish that a minimal set of transcriptional markers form representative signatures of recent experience, enabling precise decoding of recent salient experiences at the resolution of individual mice.

## Discussion

The brain creates representations of the world, encoding salient information for long-term storage to support the development of adaptive behaviors. In real time, the representation of information has been shown to be correlated with neural activity in distinct brain structures (*Bialek et al., 1991*). Powerful demonstrations of the potential to decode sensory experiences and correlates of emotional state have been made in both rodents and humans from neural activation patterns using in-vivo electrophysiology, fMRI, and other imaging techniques (*Horikawa et al., 2013*; *Santoro et al., 2017*; *Kragel et al., 2016*; *Lin et al., 2005*; *Reber et al., 2002*). In this study we demonstrate that multiplexed IEG expression data from multiple regions of the mouse brain enables the decoding of recent salient experiences with high precision. We show that beyond mere 'activity markers' for labeling neurons activated during an experience, IEG expression provides a quantitative and scalable metric, representing a neural transcriptional code for recent experience. This neural transcriptional code is defined by the combinatorial expression of marker transcripts across brain regions. Interestingly, we find components of induced transcriptional signatures that are associated with affective attributes of the experiences that are being encoded. Moreover, these IEG expression patterns are modulated following repeated administration of a stimulus of positive or negative value, suggesting a role for inducible transcription in sustaining long-term plasticity underlying the development of adaptive behavior. As this code is comprised of molecular components, it also provides a rich resource for biological insight into the processes underlying the long-term encoding of experience-dependent plasticity.

Transcriptional markers have been successfully utilized for the classification of developmental stages (*Matcovitch-Natan et al., 2016*), diseases (*Lamb, 2007*; *McKinney et al., 2010*), and many other aspects of contemporary biomedical science (*Collins and Varmus, 2015*). Here we describe the utility of transcriptional markers for classification of salient experiences characterized by diverse affective properties. While the information embedded in the expression pattern of a single gene is not sufficient, a minimal subset of transcriptional markers enable the decoding of recent experience with high accuracy. Importantly, the principles we identify likely generalize to a broader set of experiences. Furthermore, it is likely that markers we utilize in our study could be substituted by other markers genes, providing similar classification accuracy.

According to the Russell circumplex model (*Russell, 1980*; *Posner et al., 2005*), affect can be defined in two dimensions – valence and salience. Valence has been suggested to be encoded in the Amy, PFC, NAc and VTA (*Namburi et al., 2016*). Our results demonstrate that experiences of

negative valence are represented by a distinct transcriptional induction in the Amy. In contrast, experiences of positive valence induce transcription in the LCtx, NAc, DS and VTA. Moreover, we report that upon repetition, the transcriptional representation within these structures is dynamically modulated, potentially underlying long-term adaptations following positive and negative reinforcement. Taken together, our results suggest that inducible transcription is a rich resource for the identification of brain regions that encode properties of an experience, providing biological insight into the molecular processes underlying experience-dependent plasticity. It should be noted that in this study we focused our analysis on structures associated with limbic and mesolimbic system. It is highly likely that transcriptional signatures across other brain areas (as well as for other experiences) would be related to different attributes of the experience, besides affect or valence.

To explain how changes in transcription could affect future behavior, we introduce the concept of 'predictive transcriptional coding'. Predictive transcriptional coding frames inducible transcription not as a reporter of a recent event, but rather as encoding the valuation of the experience. This experience-dependent plasticity, mediated by transcription, sets the state of the network in the context of a particular experience, priming it for prospective network plasticity, and adjusting the response of the individual to the occurrence of a similar event in the future. This notion is conceptually similar to the 'reward prediction error' (*Schultz, 2010*), but is established on prolonged time scales. In this respect, transcription also serves as a 'salience filter' – defining whether an experience is significant enough to induce plasticity and worthy of encoding for long-term storage. Thus, the valuation of an experience that passes the 'salience filter' is encoded by the identity of the neural circuits recruited by the experience and the magnitude of transcription induced within them. A crucial question arising from this concept is: how is the threshold to commit to induction of transcription determined in neurons and neural networks? One possibility, worthy of future investigation, was proposed in a landmark treatise, in which the analogy of a 'genomic action potential' was drawn for mechanisms underlying inducible transcription (*Clayton, 2000*). According to this hypothesis, the threshold for commitment to transcription depends on the coincidence of glutamatergic and neuromodulatory inputs.

Our work provides a numerical definition of the imprint of recent experience, demonstrating a quantitative and predictive approach for the analysis of neural plasticity underlying adaptive behavior. Quantitative definitions of interoceptive states are expected to have implications for drug development - providing objective metrics for comprehensive characterization of the perception and valuation ascribed to an experience by individual subjects. For example, in the context of abuse liability, an objective quantitative interoceptive metric of the hedonic potential of a compound could increase standardization, reducing the reliance on variable behavioral outcomes.

While there is substantial investment being made in the development of methodologies for transcriptional profiling with deeper coverage and increasing spatial resolution, our study demonstrates that fundamental phenomena can be identified by applying simple methods with low spatial resolution and coverage. Future work, applying tools of higher resolution, could build on our observations to address additional questions – such as the spatial distribution of neuronal ensembles recruited by experience and the identity of cell types recruited by distinct experiences.

Approaches for non-invasive quantitative measurement of the encoding of experience can be envisioned, utilizing fluorescent markers of inducible transcription in combination with whole-brain imaging (*Eguchi and Yamaguchi, 2009*). New technologies are rapidly emerging for whole-brain analyses of transcription (*Renier et al., 2016*; *Sylwestrak et al., 2016*; *Ye et al., 2016*), as are strategies for comprehensive profiling of single neurons (*Citri et al., 2011*; *Lacar et al., 2016*). These technological developments, together with the novel concept we develop here, are expected to provide the foundation for a new area of neuroscience research. This discipline, of 'Behavioral Transcriptomics', will apply transcriptional analysis for investigation of intricate mechanisms of neural circuit plasticity underlying cognition. We propose that the approach of behavioral transcriptomics will provide a systems-level view of the encoding of experiences to long-term memory. One could speculate that different attributes of an experience may be mediated by activation of defined signaling pathways at different cellular locations, each inducing a component of the transcriptional program. If so, taken to its extreme, deciphering this transcriptional code will enable precise decoding of synapse-specific plasticity from quantitative analysis of inducible transcriptional markers.

## Materials and methods

### Animals

Male C57BL/6 mice aged 6–8 weeks (Harlan Laboratories, Jerusalem, Israel) served as subjects for the study. Mouse body mass ranged from 18 to 35 g, while between experimental groups in each repetition of experiments, the difference in body mass between animals did not exceed four grams. Four to five mice were housed per cage in all experiments except for sucrose consumption experiments, for which animals were single-housed. Mice were maintained in 12–12 hr light/dark cycle (0700 on/1900 off), in a temperature (20–22°C) and humidity (55 ± 10%) controlled facility. Mice received ad libitum access to water and food, with the exception of the experiment studying reinstatement of feeding, in which they were food deprived for 18 hr before reinstatement of feeding. Mice were randomly assigned to experimental groups and tested according to Latin square design. All tests were conducted during the light phase of the circadian cycle. Each experiment was performed at least twice, by independent researchers in the group, and provided similar results. All animal protocols were approved by the Institutional Animal Care and Use Committees at the Hebrew University of Jerusalem and were in accordance with the National Institutes of Health Guide for the Care and Use of Laboratory Animals. A table defining the number of mice ('n') contributing to each experiment is included as *Supplementary file 2*.

### Behavioral assays

Mice were acclimated to the animal facility for at least 2–5 days, followed by 3–4 days of experimenter handling, before the start of an experiment. Maintenance of uniform conditions across experiments and extensive handling were essential for reducing experimental variability, enabling the identification of a robust transcriptional response specifically induced by the experience being tested and minimal contamination from contextual background. **Behavioral sensitization to cocaine**. Mice were subjected to three days of intraperitoneal (i.p.) saline injections (250 microliter/injection), prior to exposure to cocaine (20 mg/kg freshly dissolved in physiological saline to 2 mg/ml and injected at a volume of 10 ml/kg; cocaine was obtained from the pharmacy at Hadassah Hospital, Jerusalem). The *acute cocaine* group received a single i.p. dose of cocaine, followed by analysis of locomotor behavior for 15 min in a video-monitored open-field arena. Animals were finally taken from their home cage and sacrificed at 1, 2 and 4 hr following the cocaine injection. The *repeated cocaine* group received five consecutive daily injections of cocaine and were studied (similar to the acute cocaine group) following the fifth cocaine injection. The *challenge cocaine* group were treated as the repeated cocaine group, and then made abstinent from cocaine for 21–22 days, following which they were challenged with cocaine and re-exposed to the open-field arena. All responses were normalized to baseline controls (time 0), which were interleaved with their peer group, but were not treated on the day of the experiment. Additional reference groups included *acute saline without habituation*, which were habituated to the open-field arena for three days after a brief period of handling, and were sacrificed 1 hr following a single injection of saline. Responses in this group were normalized to controls (time 0), which were not exposed to any saline injections. The group of *acute saline without habituation* served as a reference for the habituation of the *acute saline* group, in which animals were treated identically to the acute cocaine group (i.e. three consecutive days of habituation to saline injections in the open-field arena), but received a saline injection on the day of the experiment. Following each i.p. injection, mice were placed in an open-field arena for 20 min, during which locomotion was assayed between minutes 2 to 17. **LiCl exposure**. All mice were habituated to injections of saline and locomotor monitoring in an open-field arena for three days preceding onset of the experiment. Animals were subjected to either acute or repeated administration of LiCl (Sigma-Aldrich, St.Louis, MO, USA). In *acute LiCl* experiments, mice were administered with either a single dose of LiCl (at 150 or 250 mg/kg) or saline. In the experiments testing *repeated LiCl*, mice received LiCl (150 mg/kg) for five consecutive days, and following a 48 hr break were re-exposed to LiCl or saline. Mice were divided into four groups: a) Received saline injections for five days and were not exposed to an injection on the last day (*saline-0h*), b) Received LiCl injections for five days and were not exposed to an injection on the last day (*LiCl-0h*), c) Received saline injections for five days and were subjected to saline injection on the last day (*repeated saline*), d) Received LiCl injections for five days and were exposed to LiCl injection on the last day (*repeated*

*LiCl*). In all experiments, immediately following administration of LiCl or saline, mice were placed in video-monitored open-field arenas for 30 min. **Reinstatement of feeding**. Mice were food deprived for 18 hr before the experiment and then re-exposed to food for 1, 2 or 4 hr before they were sacrificed. Control animals (0 hr) were sacrificed immediately after the 18 hr food restriction. An additional reference group was allowed to continuously feed. **Sucrose Consumption**. Mice were single-housed for at least seven days before the experiment and habituated to the addition of a second water bottle in the cage for three days before the onset of the experiment. *Acute exposure to sucrose* was tested by habituating mice to the bottle with 10% sucrose overnight (16 hr), and 48 hr later, re-exposing the mice to a bottle with sucrose or water (control) for 1 hr. *Repeated exposure to sucrose* was tested by exposing mice to sucrose repeatedly for eight consecutive days, 2 hr each day (12:00-14:00), and after a 48 hr break, re-exposed to sucrose or water (control) for 1 hr. Mice were sacrificed 1 hr following the exposure to sucrose. Sucrose and water intake were measured as a test for sucrose preference over water. **Foot Shock**. Following habituation to the experimental setup, the mice were placed in the experimental chamber (20 × 18 cm) for three minutes, during which time, baseline freezing behavior was measured. At three minutes, each subject received three mild foot shocks (2 s, 0.7 mA) separated by 30 s interval and post-shock freezing behavior was assessed immediately thereafter for 30 s before return to the home cage. Freezing, defined as a lack of movement other than respiration, was measured using Ethovision software (Noldus, Wageningen, The Netherlands).

## Locomotor activity measurement

Locomotor activity was assessed in sound- and light-attenuated open-field chambers. Mice were placed individually in a clear, dimly lit Plexiglas box (30 × 30 × 30 cm) immediately after injection of cocaine, LiCl or saline. Activity was monitored with an overhead video camera for 20 or 30 min (in cocaine sensitization and LiCl experiments respectively) using Ethovision software (Noldus, Wageningen, The Netherlands).

## Dissections

Performed as previously described (*Turm et al., 2014*). Mice were deeply anesthetized with Isoflurane (Piramal Critical Care, Bethlehem, PA, USA) and euthanized by cervical dislocation, followed by rapid decapitation and harvesting of brains into ice cold artificial cerebrospinal fluid (ACSF) solution (204 mM sucrose, 26 mM NaHCO3, 10 mM glucose, 2.5 mM KCl, 1 mM NaH$_2$PO$_4$, 4 mM MgSO$_4$ and 1 mM CaCl$_2$; all from Sigma-Aldrich, St. Louis, MO). Coronal slices (400 µm) were cut on a vibrating microtome 7000 smz2 (Camden Instruments, Loughborough, UK) in ice-cold artificial cerebrospinal fluid (ACSF). Brain regions [Limbic cortex (LCtx), Nucleus Accumbens (NAc), Dorsal Striatum (DS), Amygdala (Amy), Lateral Hypothalamus (LH), Hippocampus (Hipp) and Ventral Tegmental Area (VTA)] were dissected from relevant slices under a stereoscope (Olympus, Shinjuku, Tokyo, Japan). Samples of LCtx, NAc, DS, Amy, LH AND Hipp were obtained from 2* 400 µm thick sections, while VTA, was obtained from 2* 200 µm thick sections (*Figure 1—figure supplement 1*). All of the steps were performed in strictly cold conditions (~4°C) and care was taken to avoid warming of the tissue sections or the ASCF at all times. The tissue pieces were immediately submerged in Tri-Reagent (Sigma-Aldrich, St.Louis, MO) and stored at −80°C until processing for RNA extraction.

## Marker selection, RNA extraction, qPCR and microfluidic qPCR

The strategy for marker selection consisted of three steps. The initial list of candidate IEGs was compiled from a whole-genome microarray analysis of transcriptional dynamics induced by cocaine experiences in the nucleus accumbens (Illumina MouseRef-8 v2 Expression BeadChip microarrays; data not shown), as well as a survey of literature and databases pertaining to IEG expression. qPCR primer probes were developed for 212 genes and primer efficiency was tested, resulting in selection of 152 optimal primer pairs. Differential expression of the shortlisted IEGs was then tested on samples from multiple brain structures, dissected from mice following cocaine and LiCl experiences, utilizing microfluidic qPCR arrays. Genes that displayed at least 1.25-fold induction in any measurement were shortlisted, resulting in a list of 78 genes. The next round of feature selection involved ranking genes based on their frequency of induction and variance. For ranking based on frequency of induction, we counted the number of times each gene was induced above a threshold

of two-fold induction across the different brain structures (LCtx, NAc, DS, Amy, LH and Hipp) in the cocaine (acute, repeated and challenge) and LiCl (acute and repeated) conditions (i.e. induction in six structures*five experiences = #/30). In addition, we ranked genes in inverse order of average variance ($S^2$) of their induction across structures. The five genes that were induced most consistently (combined highest ranking in frequency and lowest in variance) were selected for further investigation. The ranking of these genes was as follows: *Arc* (#=22/30, $S^2$ = 2.9), *Egr2* (#=21/30, $S^2$ = 2.8), Egr4 (#=18/30, $S^2$ = 1.53), *Fos* (#=14/30, $S^2$ = 0.43), *Fosb* (#=11, $S^2$ = 0.6). Thus, criteria for marker selection were orthogonal to the tested hypothesis, supporting unbiased analysis.

RNA extraction was performed strictly in cold RNase-free conditions. Tissue was homogenized using a 25G needle attached to a 1 ml syringe or using TissueLyser LT (Qiagen, Redwood city, CA, USA). The homogenate was centrifuged at high speed (15 k g for 10 min) and the supernatant was mixed with chloroform (Bio-Lab, Jerusalem, Israel) by vigorous shaking and centrifuged (15 k g for 15 min) to separate the RNA from other nucleic acids and proteins. Isopropanol (J. T. Baker, Center Valley, PA) and glycogen (Roche, Basel, Switzerland) were added to the aqueous layer and samples were placed either at −20°C for 24 hr or at −80°C for 1 hr (producing comparable results). The samples were centrifuged at high speed (15 k g for fifteen min) for the precipitation of the RNA. The RNA was then washed in 75% ethanol (J. T. Baker, Center Valley, PA) by centrifugation (12 k g for five min), dried and dissolved in ultrapure RNase free water (Biological Industries, Kibbutz Beit Haemek, Israel). RNA concentration was measured with a NanoDrop 2000c spectrophotometer (Thermo, Wilmington, DE) and random-primed cDNA was prepared from 100 to 300 ng of RNA, with use of a High Capacity cDNA Reverse Transcription Kit (Applied Biosystems, Foster city, CA), following manufacturer guidelines.

cDNA was processed for qPCR analysis using qPCR primer pairs (IDTDNA, Coralville, IA) and SYBR Green in a Light-cycler 480 Real Time PCR Instrument (Roche Light Cycler*480 SYBR Green I Master, Roche, Basel, Switzerland) according to manufacturer guidelines. Relative levels of gene expression (ΔCt) were obtained by normalizing gene expression to a housekeeping gene (GAPDH). Fold induction was calculated using the ΔΔCt method, normalizing experimental groups to the average of a relevant control group.

Microfluidic qPCR, querying 96 samples against 96 sets of qPCR probes was performed utilizing Fluidigm Biomark Dynamic IFC (integrated fluidic circuit) Arrays (Fluidigm Corp, South San Francisco, CA). Briefly, samples are subjected to targeted preamplification to enrich for specific gene products, which were then assayed with dynamic array fluidic microchips. Sample preparation was performed according to previously published protocols (*Turm et al., 2014*). Targeted pre-amplification (STA) was achieved by mixing samples with a set of diluted primer pairs in TaqMan PreAmp Mastermix (Applied Biosystems; Foster City, CA, USA) followed by 10 min of denaturation at 95°C and 14 cycles of amplification (cycles of 95°C for 15 s and 60°C for 4 min). Primers were then eliminated by use of ExoI exonuclease (NEB; Ipswich, MA), placed in a thermal cycler at 37°C for 30 min and then at 80°C for 15 min. Samples were then loaded onto a primed dynamic array for qPCR in a specialized thermal cycler [Fluidigm Biomark; Thermal mixing: 70°C for 40 min, 60°C for 30 s, 95°C denaturation for 60 s, followed by 40 cycles of PCR (96°C for 5 s, 60°C for 20 s)]. For data analysis, a reference set of genes was identified, whose expression remained constant across all experimental conditions (*Dkk3, Tagln3, Gars, Scrn1, Rpl36al, Mcfd2, Psma7 and Hpcla4*). In order to reduce the potential for introduction of experimental error by normalization to a single gene, a 'global-normalization' Ct value was created for each sample from the average Ct values of the genes within the reference set. Fold induction was calculated using the ΔΔCt method, normalizing each gene in a sample to the global-normalization value (ΔCt), followed by normalization of the experimental groups to the average of their relevant control group.

## Data analysis

All data are presented as mean ± standard error (s.e.m.). Data were analyzed using one-way or two-way analysis of variance (ANOVA), as appropriate. Tukey or Dunnett test was used for post hoc analyses of significant ANOVAs to correct for multiple comparisons. Differences were considered significant at the level of $p < 0.05$. Statistical analysis was performed, and bar graphs and line graphs were created, with Prism 6.0 (GraphPad, San Diego, CA). Heat maps were created in MATLAB R2012a (Mathworks, Natick, MA). Radar plots were created in Origin 6.0 (Originlab, Northampton, MA).

Codes were written in MATLAB R2015b (MathWorks, Natick, MA) and confusion matrices, randomization plots were created in Python using the Matplotlib library (http://matplotlib.org).

## Computational analyses

The analysis was performed on data obtained from 54 mice, each of which experienced one of the experiences (acute, repeated or challenge cocaine, acute and repeated sucrose, reinstatement of feeding, acute and repeated LiCl and foot shock and no-shock controls exposed to the same environment). Each mouse was represented by a vector of twenty-five features [corresponding to the induction of five genes (*Arc, Egr2, Egr4, Fos* and *Fosb*) across five structures [limbic cortex (LCtx), nucleus accumbens (NAc), dorsal striatum (DS), amygdala (Amy) and lateral hypothalamus (LH)]. Each gene-structure combination was defined as a 'feature'.

## Supervised classification

The classifier used was k-Nearest Neighbors (KNN), with k = 1 over the Euclidean space, unless otherwise stated. This approach was selected based on the observation that the transcriptional response of mice within an experience group formed unique clusters. We evaluated the performance of our classification by a leave-one-out method. In this approach, we iterated over each sample in our training set and classification was performed given the rest of the training set. Visualization of the accuracy of classification was performed using a confusion matrix, which conveys both mean precision and mean recall of each condition classified.

## Feature selection

Feature selections were performed using Random k-Nearest Neighbors (RKNN) (*Li et al., 2011*) or Breiman Random Forest (RF) (*Breiman, 2001*) algorithm. For RKNN, the contribution of each feature for classification of individual experiences was called *support*. We chose large (n = 1e$^6$), random subsets of the twenty five available features in varying sizes (between one and twenty five). For each such subset we trained a classifier. Each feature *f* appeared in some KNN classifiers, for example, set C(*f*) of size *M*, where *M* is the multiplicity of *f*. In turn, each classifier *c* ∈ C(*f*) is an evaluator of its *m* features. We defined the *support* of a feature *f* as the mean accuracy of all the classifiers in C(f). Namely:

$$support(f) = \frac{\sum_{c \epsilon C(f)} accuracy(c)}{M}$$

To further examine the effect of feature set sizes on classification performance we evaluated the classification accuracy of different subset sizes in the following manner: for each case, we chose the n features which were ranked the highest in their support, and evaluated the KNN classifier trained with those features only.

For classification using Random Forest (RF), we used the Breiman random forest algorithm (*Breiman, 2001*; ), according to which a large number (n = 1e$^5$) of decision trees were built, where each tree used a varying number of features (between 1 and 25). Bifurcations were chosen according to modified Gini gain. For each feature, we averaged over the decrease in the Gini gain (MDG) (*Han et al., 2016*) over the ensemble of decision trees. The selected features were then evaluated using a regularized (pruned) decision tree with a maximal depth of 4, using a k-cross validation process with k = 10, with the constraint of a minimum categorization of 3 animals per group. The decision tree was constructed using the CART decision tree construction algorithm (*Breiman et al., 1984*) (*Figure 3—figure supplement 2E–H*).

## Decision tree

To provide an example of a descriptive classifier, we created a decision tree using the CART algorithm with Information Gain (*Ben-David and Shalev-Shwartz, 2014*). No constraints were applied while building this tree.

## Randomization

Considering the limited size of our dataset, we wanted to ensure that the classifier was not over fitted to our training set S. For this purpose, we produced a large number N (N = 1e$^5$) of permuted

versions of our training set (s$_i$, ...s$_N$), and created KNN or decision tree as the classifiers in the same way as for the original data. The permutation was performed by shuffling the association of individual mice with experiences. For each such permuted training set we trained a classifier and evaluated its classification accuracy (leave-one-out, see previous description). We calculated the empirical p value (p$<1e^{-5}$ for both conditions) for the classification accuracy on our original training set in the following manner:

$$p - val = \frac{1}{N}\sum_{i=1}^{N} acc_{(S_i)}(S)$$

## Data and code availability

The data sets generated during the current study, as well as the code used for analysis have all been uploaded as supplementary material (*supplementary file 1–4*, *source code 1–11*).

## Acknowledgements

This work was funded by grants to AC from the Israel Science Foundation (393/12 and 2341/15), ISF Center of Research Excellence on 'Chromatin and RNA in Gene Regulation' (1796/12), EU Marie Curie (PCIG13-GA-2013–618201), the Brain and Behavior Foundation (NARSAD 18795), the German-Israel Foundation (2299–2291.1/2011), the Binational Israel-USA Foundation (2011266), the Milton Rosenbaum Endowment Fund for Research in Psychiatry, the Canadian Institute for Advanced Research, and contributions from Mr. Jaime Cohen (Mexico City) and the Stewart Resnick Foundation (Los Angeles). BIJ was funded by the Shimon Peres Fellowship from the Edmond and Lily Safra Center for Brain Sciences and the Lady Davis Fellowship Trust. Rob Malenka's generosity in enabling preliminary studies to be performed in his laboratory is highly appreciated. We thank Hermona Soreq, Inbal Goshen, Mickey London, Sagiv Shifman, Zhiping Pang and members of the Citri lab for constructive criticism of the manuscript.

## Additional information

### Funding

| Funder | Grant reference number | Author |
|---|---|---|
| Shimon Peres Postdoctoral Award | Postdoctoral stipend | Bogna Marta Ignatowska-Jankowska |
| ELSC Postdoctoral Award | Postdoctoral stipend | Bogna Marta Ignatowska-Jankowska |
| Lady Davis Fellowship Trust, Hebrew University of Jerusalem | Postdoctoral stipend | Bogna Marta Ignatowska-Jankowska |
| Israel Science Foundation | Personal Grant 393/12 & I-CORE 1796/12 | Ami Citri |
| German-Israeli Foundation for Scientific Research and Development | Young Investigator Award 2299-2291.1./2011 | Ami Citri |
| Brain and Behavior Research Foundation | Young Investigator Award #18795 | Ami Citri |
| Canadian Institute for Advanced Research | Research Support | Ami Citri |
| Binational United-States Israel Research Foundation | Research Grant #2011266 | Ami Citri |
| Milton Rosenbaum Research Foundation | Research Grant | Ami Citri |
| National Institute for Psychobiology in Israel, Hebrew University of Jerusalem | Research Grant 109-15-16 | Ami Citri |

| Israel Science Foundation | 2341/1 | Ami Citri |
| H2020 Marie Skłodowska-Curie Actions | PCIG13-GA-2013–61820 | Ami Citri |
| Stewart Resnick Foundation | | Ami Citri |

The funders had no role in study design, data collection and interpretation, or the decision to submit the work for publication.

## Author contributions
Diptendu Mukherjee, Bogna Marta Ignatowska-Jankowska, Conceptualization, Data curation, Formal analysis, Validation, Investigation, Visualization, Methodology, Writing—original draft, Writing—review and editing; Eyal Itskovits, Conceptualization, Software, Formal analysis, Methodology; Ben Jerry Gonzales, Investigation, Methodology, Writing—review and editing; Hagit Turm, Validation, Investigation, Methodology, Project administration; Liz Izakson, Doron Haritan, Noa Bleistein, Chen Cohen, Validation, Investigation, Methodology; Ido Amit, Tal Shay, Resources, Methodology; Brad Grueter, Resources, Investigation, Methodology; Alon Zaslaver, Software, Formal analysis, Supervision; Ami Citri, Conceptualization, Resources, Formal analysis, Supervision, Funding acquisition, Visualization, Methodology, Writing—original draft, Project administration, Writing—review and editing

## Author ORCIDs
Bogna Marta Ignatowska-Jankowska (iD) http://orcid.org/0000-0002-8427-1128
Brad Grueter (iD) http://orcid.org/0000-0002-4224-3866
Ami Citri (iD) http://orcid.org/0000-0002-9914-0278

## Ethics
Animal experimentation: This study was performed in strict accordance with the recommendations in the Guide for the Care and Use of Laboratory Animals of the National Institutes of Health. All of the animals were handled according to approved institutional animal care and use committee (IACUC) protocols (#NS-13-13895-3; NS-15-14668-3; NS-14-14088-3; NS-15-14312-3; NS-15-14348-3) of the Hebrew University of Jerusalem. The protocol was approved by the Committee on the Ethics of Animal Experiments of the Hebrew University. Every effort was made to minimize suffering.

## Decision letter and Author response
Decision letter https://doi.org/10.7554/eLife.31220.036
Author response https://doi.org/10.7554/eLife.31220.037

# Additional files

## Supplementary files
• Source Code 1. Classfication Tests_averageGenes
DOI: https://doi.org/10.7554/eLife.31220.019

• Source Code 2. Classfication Tests_averageRegions
DOI: https://doi.org/10.7554/eLife.31220.020

• Source Code 3. Classfication Tests_pcaOverRegions
DOI: https://doi.org/10.7554/eLife.31220.021

• Source Code 4. ClassificationAndConfusionMatrices_confMat
DOI: https://doi.org/10.7554/eLife.31220.022

• Source Code 5. Classification And Confusion Matrices_knnEvaluate
DOI: https://doi.org/10.7554/eLife.31220.023

• Source Code 6. Feature Selection RKNN_find Features
DOI: https://doi.org/10.7554/eLife.31220.024

• Source Code 7. Feature Selection RKNN_find Interesting Genes
DOI: https://doi.org/10.7554/eLife.31220.025

- Source Code 8. Features Selection Random Forest_DecisionTree
DOI: https://doi.org/10.7554/eLife.31220.026

- Source Code 9. Linear Projections_RegionsScatter
DOI: https://doi.org/10.7554/eLife.31220.027

- Source Code 10. Linear Projections
DOI: https://doi.org/10.7554/eLife.31220.028

- Source Code 11. Randomization_permTest
DOI: https://doi.org/10.7554/eLife.31220.029

- Supplementary file 1. Primer sequences and efficiency calculations.
DOI: https://doi.org/10.7554/eLife.31220.030

- Supplementary file 2. Raw data.
DOI: https://doi.org/10.7554/eLife.31220.031

- Supplementary file 3. Statistics.
DOI: https://doi.org/10.7554/eLife.31220.032

- Supplementary file 4. Animal numbers
DOI: https://doi.org/10.7554/eLife.31220.033

- Transparent reporting form
DOI: https://doi.org/10.7554/eLife.31220.034

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
