## [Decision Letter]

Thank you for submitting your article "Salient Experiences are Represented by Unique Transcriptional Signatures in the Brain" for consideration by *eLife*. Your article has been favorably evaluated by Aviv Regev (Senior Editor) and three reviewers, one of whom, Sacha Nelson, is a member of our Board of Reviewing Editors. The following individual involved in review of your submission has agreed to reveal their identity: Pavel Osten (Reviewer #2).

The reviewers have discussed the reviews with one another and the Reviewing Editor has drafted this decision to help you prepare a revised submission.

Summary:

The authors use real-time PCR from dissected brain regions to assess the immediate early gene (IEG) transcriptional responses to a variety of rewarding and aversive experiences including drugs of abuse, feeding, foot shock and gastric distress. Results show specific transcriptional signatures for each experience, enabling decoding of the experience. Moreover, transcriptional codes appeared to primarily reflect the valence of the event, with experiences of neutral or negative valence showing opposing patterns of transcriptional activity. This is an interesting and novel approach akin to efforts to decode experience from brain imaging and neural recordings.

Essential revisions:

1) Increasing the rigor of Feature Selection: There is a fundamental issue with the number of features considered in the analyses and how they were selected from what was measured. Initially, 152 IEGs are measured in 7 brain areas (=1064 features). Then, the number of IEGs is reduced to 78 (>1.25-fold induction). From these, 5 are selected (*Arc, Egr2, Egr4, Fos* and *Fosb*), but this step is not well described or justified. Potentially, some of the reduction could be attributed to a priori assumptions based on the literature, but these should be orthogonal to the tested hypotheses (see below) and need to be justified.

In addition, some analyses only consider a smaller number of brain areas, without clear justification. All of these choices can be extremely impactful and may strongly affect the results and ultimately the conclusions drawn from the data. Most importantly, criteria for these choices need to be orthogonal to the tested hypothesis (they cannot depend on differences in IEGs between experiences). If feature selection is non-independent, subsequent analyses will be biased and produce invalid results. The authors need to describe what criteria were used for feature selection (regions and IEGs) and whether these criteria were orthogonal to comparisons among individual experiences. For example, the formal feature selection procedure conducted for results reported in Figure 3 is non-independent, resulting in biased (invalid) classification accuracies. The problem is that the same data were used to select features (estimate "support") and also to evaluate classifier performance, which is circular. Feature selection must be based on nested cross-validation within the training data only. For instance, the algorithm should start with all 1064 features (or 78 x 7 = 546 features) and use nested cross-validation to select a feature set that is then used to determine classifier accuracy in the left-out test data.

2) Controlling for multiple comparisons – this is required for all tests.

3) Statistical testing is needed for many results presented in the manuscript. No statistical tests are reported for any of the results summarized in the sentence "The representation of rewarding experiences are characterized by robust transcriptional induction in the LCtx, NAc, DS, and VTA, while the representations of aversive experiences are dominated by transcriptional induction in the Amy". There are also no tests for whether transcriptional signatures of different experiences with the same or different valence are positively or negatively correlated. All descriptive statements should be backed by appropriate statistical tests within the manuscript.

4) The authors contrast the idea of a "transcriptional code" with the idea of IEG expression as simply "molecular markers for labelling neuronal populations that undergo plastic changes." The issue of the degree to which differences in which genes are transcribed vs. where in the brain they are transcribed is not satisfyingly analyzed. Encoding that matches different experiences to different transcripts would be a transcriptional code in the sense that many might assume from use of the term. On the other hand, encoding that matches different experiences to different brain regions would be quite akin to the "molecular marker" model the authors wish to reject. In between these two extremes, and probably closer to the data is the view that different experiences activate different brain regions, but that different brain regions also have different preferred mixtures of IEGs to activate. Gene-structure pairs are treated as features, but the relative degree to which this is a "spatial" code across brain regions vs. a genetic code across genes is unclear. The authors should attempt to separate these two contributing factors to more precisely specify what kind of "neural code embedded in transcription" they are talking about. It looks from Figure 2, for example, like there is a strong "shape similarity" across experiences within a region. This would seem to imply that relative activation of the 4 genes tested is more a function of the region than of the experience (e.g. LH and VTA have relatively little activation of *Arc*). On the other hand, overall magnitude is more related to the interaction between the experience and the structure (cocaine for VTA and DS; foot shock for hipp).

5) The current findings suggest that transcriptional signatures for individual experiences are primarily driven by salience and valence. However, this could simply be a consequence of the (reward-related) brain regions considered here. It is possible that transcriptional signatures in other brain areas are not related to valence, and this should be discussed.

6) Title: it is convention for *eLife* papers to include some reference to the preparation used. This could be achieved by including the word "rodent" or at least "mammalian" before the word "brain" in the title.

---

## [Author Response]

Essential revisions:1) Increasing the rigor of Feature Selection: There is a fundamental issue with the number of features considered in the analyses and how they were selected from what was measured. Initially, 152 IEGs are measured in 7 brain areas (=1064 features). Then, the number of IEGs is reduced to 78 (>1.25-fold induction). From these, 5 are selected (Arc, Egr2, Egr4, Fos and Fosb), but this step is not well described or justified. Potentially, some of the reduction could be attributed to a priori assumptions based on the literature, but these should be orthogonal to the tested hypotheses (see below) and need to be justified.In addition, some analyses only consider a smaller number of brain areas, without clear justification. All of these choices can be extremely impactful and may strongly affect the results and ultimately the conclusions drawn from the data. Most importantly, criteria for these choices need to be orthogonal to the tested hypothesis (they cannot depend on differences in IEGs between experiences). If feature selection is non-independent, subsequent analyses will be biased and produce invalid results. The authors need to describe what criteria were used for feature selection (regions and IEGs) and whether these criteria were orthogonal to comparisons among individual experiences. For example, the formal feature selection procedure conducted for results reported in Figure 3 is non-independent, resulting in biased (invalid) classification accuracies. The problem is that the same data were used to select features (estimate "support") and also to evaluate classifier performance, which is circular. Feature selection must be based on nested cross-validation within the training data only. For instance, the algorithm should start with all 1064 features (or 78 x 7 = 546 features) and use nested cross-validation to select a feature set that is then used to determine classifier accuracy in the left-out test data.

We thank the reviewers for these comments, which are important and indeed require explicit clarification. Three separate issues arise, to which we relate: (1) The basis for selection of the 5 genes used for analyses throughout the paper. (2) The basis for selection of brain structures. (3) Feature selection vs classifier performance in Figure 3 – ensuring that there was no circular logic.

1) Clarification for the selection of 5 genes:

We have now clarified and elaborated on the process of marker gene selection in the text (see below). In order to assist this clarification, we have also reordered the figures, and now include the analysis of LiCl in Figure 1, and add a comparison of genes induced by cocaine and LiCl, which assists in clarifying the process of feature selection. Furthermore, the text describing the process of selection of 5 genes for analyses is elaborated both in the Results as well as in the Materials and methods sections. Importantly, the criteria for selection of the 5 marker genes were orthogonal to, and independent of, the tested hypothesis, supporting unbiased analysis.

In the body of the Results section, we define the process as follows: “We analyzed the transcription induced at 0, 1, 2, 4 hours following each of these cocaine experiences across 6 brain structures (LCtx, NAc, DS, Amy, LH, and Hipp; Figure 1—figure supplement 1).[…] The 5 genes with combined highest ranking in frequency of appearance and lowest variance in expression were selected) for further analysis (*Arc, Egr2, Egr4, Fos* and *Fosb*; Figure 1).”

Within the Materials and methods section, we further elaborate on the process of marker selection, as follows: “The strategy for marker selection consisted of three steps.[…] Thus, criteria for marker selection were orthogonal to the tested hypothesis, supporting unbiased analysis.”

2) Selection of 5 brain structures:

We chose to focus on 5 brain structures for analysis (LCtx, NAc, DS, Amy and LH). The primary reason for utilizing these structures for analysis was that these were the structures for which we obtained samples across all experiences. Furthermore, hippocampal samples demonstrated very little transcriptional induction, and samples from the VTA were the most difficult to reliably obtain, due to the small size and irregular shape of this structure. In the original manuscript we performed a classification of cocaine experiences based on 3 structures (LCtx, NAc and DS; originally Figure 1). This analysis has been removed from the revised version of the manuscript, and replaced with a decoding of the recent experience of mice following cocaine and LiCl experiences, using the data from all 25 features (5 structures * 5 genes; Figure 1).

3) Feature Selection vs. classifier performance:

Regarding the feature selection described in Figure 3 – ‘support’ was calculated with the primary objective of identifying the features contributing most to the classification, but the classifier does not depend on feature selection. We have now clarified this in the revised manuscript, in which we present the decoding based on all 25 features (with no selection) in the main figure (Figure 3). The analysis of support and the identification of the most indicative features have been moved to the supplementary information (Figure 3—figure supplement 2). In addition, we have utilized an independent approach to feature selection (Breiman Random Forest), which provided a largely overlapping set of the most informative features (both approaches identify the same top 12 features, and overlap in 6 of the top 8 features). It is important to stress that the issue of feature selection is not cardinal in any way for the theme of this paper. The KNN classifier does not require feature selection to enable powerful classification of the experiences, and it is our conviction, as we have written in the Discussion, that the marker genes we have utilized could largely be substituted for a different set, without a significant loss in classification efficiency: “Furthermore, it is likely that markers we utilize in our study could be substituted by other markers genes, providing similar classification accuracy.”.

This text now reads: “Finally, we tested our capacity to decode the recent experience of mice on the full complement of experiences studied in this project. […] An independent approach for feature selection (Breiman Random Forest^25^) identified a largely overlapping set of features, with the top 10 features supporting a classification accuracy of 94.4% (p<1e^-5^; Figure 3—figure supplement 2).”

Finally, the reviewers propose that we perform feature selection based on nested cross-validation within the training data using 1064 or 546 features. This would be wonderful to implement, but is not possible in our data – as we have not acquired the full dataset of 152 genes for all the experiences studied. We acquired the 78 gene dataset for a *subset* of the mice that underwent cocaine, LiCl and saline experiences, while the majority of mice and experiences were analyzed with the 5 gene set defined based on the results of the analysis of LiCl and cocaine experiences. Therefore, the process we developed for selecting genes for analysis was the least biased process we could perform. This process, of selecting features based on high fold induction and low variance, is orthogonal to the hypothesis that recent experiences can be decoded from a transcriptional signature and enables unbiased analysis. Furthermore, the results of our study do not depend, to any significant extent, on feature selection.

2) Controlling for multiple comparisons – this is required for all tests.

Indeed, we have controlled for multiple comparisons, as defined in the Materials and methods section: “Data analysis. All data are presented as mean ± standard error (s.e.m.). […] Codes were written in MATLAB R2015b (MathWorks, Natick, MA) and confusion matrices, randomization plots were created in Python using the Matplotlib library (http://matplotlib.org).”

3) Statistical testing is needed for many results presented in the manuscript. No statistical tests are reported for any of the results summarized in the sentence "The representation of rewarding experiences are characterized by robust transcriptional induction in the LCtx, NAc, DS, and VTA, while the representations of aversive experiences are dominated by transcriptional induction in the Amy". There are also no tests for whether transcriptional signatures of different experiences with the same or different valence are positively or negatively correlated. All descriptive statements should be backed by appropriate statistical tests within the manuscript.

We have performed a thorough statistical analysis of our data, and include a supplementary table describing this analysis (Supplementary file 3 – Statistics), to which we refer throughout the manuscript. We have also included a much more detailed description of statistical significance throughout the Results section.

4) The authors contrast the idea of a "transcriptional code" with the idea of IEG expression as simply "molecular markers for labelling neuronal populations that undergo plastic changes." The issue of the degree to which differences in which genes are transcribed vs. where in the brain they are transcribed is not satisfyingly analyzed. Encoding that matches different experiences to different transcripts would be a transcriptional code in the sense that many might assume from use of the term. On the other hand, encoding that matches different experiences to different brain regions would be quite akin to the "molecular marker" model the authors wish to reject. In between these two extremes, and probably closer to the data is the view that different experiences activate different brain regions, but that different brain regions also have different preferred mixtures of IEGs to activate. Gene-structure pairs are treated as features, but the relative degree to which this is a "spatial" code across brain regions vs. a genetic code across genes is unclear. The authors should attempt to separate these two contributing factors to more precisely specify what kind of "neural code embedded in transcription" they are talking about. It looks from Figure 2, for example, like there is a strong "shape similarity" across experiences within a region. This would seem to imply that relative activation of the 4 genes tested is more a function of the region than of the experience (e.g. LH and VTA have relatively little activation of Arc). On the other hand, overall magnitude is more related to the interaction between the experience and the structure (cocaine for VTA and DS; foot shock for hipp).

We thank the reviewers for this insightful comment. We have performed additional analysis, addressing the decoding capacity when we relate to each structure independently (5 genes in each individual structure), genes alone (losing the reference to structures by averaging across structures) or individual genes (relating to each gene separately). This analysis reveals that while individual genes (specifically *Egr2* and *Fos*) are strong classifiers on their own (correctly classifying 67-70% of the mice) the predictive power improves substantially when relating to multiple features, comprising of both genes and structures.

As defined in the text: “To further test this hypothesis, we ran a number of permutations. […]. Taken together, while we find that measurement of the expression of individual genes, such as *Fos* and *Egr2*, across the 5 brain structures can support classification (67%, 70% respectively), the prediction is significantly improved by the measurement of multiple features (Figure 3).”

5) The current findings suggest that transcriptional signatures for individual experiences are primarily driven by salience and valence. However, this could simply be a consequence of the (reward-related) brain regions considered here. It is possible that transcriptional signatures in other brain areas are not related to valence, and this should be discussed.

We accept this comment, and have included a sentence to this avail in the Discussion. This sentence reads: “It should be noted that in this study we focused our analysis on structures associated with limbic and mesolimbic system. It is highly likely that transcriptional signatures across other brain areas (as well as for other experiences) would be related to different attributes of the experience, besides affect or valence.”

6) Title: it is convention for eLife papers to include some reference to the preparation used. This could be achieved by including the word "rodent" or at least "mammalian" before the word "brain" in the title.

We have included the word “mouse” in the title.